# Italian cross-cultural adaptation of the Quality of Communication questionnaire and the 4-item advance care planning engagement questionnaire

**Ludovica De Panfilis**[1], **Simone Veronese**[2], **Marta Perin**[1,3], **Marta Cascioli**[4], **Mariangela Farinotti**[5], **Paola Kruger**[6], **Roberta M. Zagarella**[1,7], **J. R. Curtis**[8,9], **Rebecca L. Sudore**[10,11], **Elizabeth L. Nielsen**[8,9], **Ruth A. Engelberg**[8,9], **Andrea Giordano**[5]*, **Alessandra Solari**[5], on behalf of the ConCure-SM project[¶]

**1** Bioethics Unit, Azienda USL-IRCCS di Reggio Emilia, Reggio Emilia, Italy, **2** Fondazione FARO, Turin, Italy, **3** PhD Program in Clinical and Experimental Medicine, University of Modena and Reggio Emilia, Modena, Italy, **4** Hospice 'La Torre sul Colle', Azienda USL Umbria 2, Spoleto, Italy, **5** Unit of Neuroepidemiology, Fondazione IRCCS Istituto Neurologico Carlo Besta, Milan, Italy, **6** EUPATI Fellow (European Patients Academy for Therapeutic Innovation) Italy, Rome, Italy, **7** National Research Council (CNR), Interdepartmental Center for Research Ethics and Integrity (CID Ethics), Rome, Italy, **8** Cambia Palliative Care Center of Excellence at UW Medicine, University of Washington, Seattle, Washington, United States of America, **9** Division of Pulmonary, Critical Care, and Sleep Medicine, Department of Medicine, University of Washington, Seattle, Washington, United States of America, **10** Division of Geriatrics, School of Medicine, University of California San Francisco, San Francisco, San Francisco, California, United States of America, **11** San Francisco Veterans Affairs Health Care System, San Francisco, California, United States of America

¶ Membership of the ConCure-SM project is listed in the Acknowledgments.
* andrea.giordano@istituto-besta.it

**Data Availability Statement:** All relevant data are within the paper and its Supporting information files.

## Abstract

### Background

Advance care planning (ACP) is influenced by several factors (e.g., patient's readiness to engage, clinician's skills, and the cultural environment). Availability of reliable and valid self-reported measures of the ACP domains is crucial, including cross-cultural equivalence.

### Aim

To culturally adapt into Italian the 19-item Quality of Communication (QOC) and the 4-item ACP Engagement (4-item ACP-E) questionnaires.

### Methods

We translated and culturally adapted the two questionnaires and produced a significant other (SO) version of the QOC (QOC-SO). Each questionnaire was field tested via cognitive interviews with users: nine patients (QOC, 4-item ACP-E) and three SOs (QOC-SO) enrolled at three palliative care services.

**Funding:** This research was funded by Fondazione Italiana Sclerosi Multipla (FISM; aism.fism.it), grant no. 2020/R-Multi/024 to A.S, and partially supported by the Italian Ministry of Health (RRC). R.L.S. is funded in part by the National Institute on Aging, National Institutes of Health (K24AG054415).The funders had no role in study design, data collection and analysis, decision to publish, or preparation of the manuscript.

**Competing interests:** A.S. reports grants from the Italian Multiple Sclerosis Foundation (FISM) and the European Academy of Neurology, during the conduct of the study; she serves as board member for Merck Serono, and received personal fees from Almirall and Merck Serono, outside the submitted work. This does not alter our adherence to PLOS ONE policies on sharing data and materials.

## Results

We made minor changes to 5/19 QOC items, to improve clarity and internal consistency; we changed the response option 'didn't do' into 'not applicable'. Finally, we slightly revised the QOC to adapt it to the paper/electronic format. QOC debriefing revealed that the section on end of life was emotionally challenging for both patients and SOs. We simplified the 4-item ACP-E layout, added a sentence in the introduction, and revised the wording of one item, to improve coherence with the Italian ACP legislation. ACP-E debriefing did not reveal any major issue.

## Conclusions

Results were satisfactory in terms of semantic, conceptual and normative equivalence of both questionnaires. Acceptability was satisfactory for the 4-item ACP-E, while findings of the QOC cognitive debriefing informed a major amendment of a pilot trial protocol on ACP in multiple sclerosis (ConCure-SM): use of the interviewer version only, in an adaptive form. Psychometric testing of both questionnaires on a large, independent sample will follow.

## Introduction

Advance care planning (ACP) is a process that "enables individuals who have decisional capacity to identify their values, to reflect upon the meanings and consequences of serious illness scenarios, to define goals and preferences for future medical treatment and care, and to discuss these with family and healthcare professionals" [1]. Consistent with the shared decision making model [2], ACP helps the patients to identify their own personal values and goals, understand their health status, and the treatment and healthcare options available. Moreover, it encourages discussion around end-of-life (EOL) care, a subject that is generally not considered part of healthcare planning, and one that is often avoided by both patients and health professionals. ACP involves many behaviors, including values identification, communication, and documentation; it is influenced by many factors, such as the patient's readiness to engage, the clinician's skills, the disease trajectory, and the cultural and logistic environment [3, 4]. Despite having been regulated for more than five years (Law 219/2017), ACP implementation in Italy remains negligible. In contrast, a recent survey showed that 88% (1752/2000) of Italian citizens considered the Law 219/2017 as quite or very important, and 76% had a positive attitude towards making/registering advance directives or ACP [5].

A multidisciplinary Delphi panel agreed on categorizing ACP outcomes into five domains: process (e.g. readiness to engage in ACP, prognostic awareness); action (e.g. decision on a surrogate, documentation of values and care preferences); quality of care (e.g. satisfaction with decision making); health status (e.g. mood symptoms, quality of life); and healthcare utilization (e.g. hospitalizations) [6]. Developing and sharing (self-reported) measures of the ACP domains that are acceptable, reliable, and valid, is crucial. Equally important is having these scales available in different languages, to allow consistent use of these instruments in different countries and cultures, for clinical and research purposes. Their availability in different languages is key for increasing equity of care, for the development of international research networks, and ultimately for strengthening research in this field.

ConCure-SM is an ongoing, multicenter project aimed to set up and evaluate the efficacy of an ACP intervention for multiple sclerosis patients in Italy. The intervention consists of a

healthcare professional training program in shared decision-making and ACP, and use of a booklet during the ACP conversation. A range of measures are collected in the pilot/feasibility trial inscribed in the project (trial registration number: ISRCTN48527663) in order to capture the full process of ACP and to assess whether the intervention has any effect on completion of an advance care plan document (primary outcome measure), congruence in treatment preferences between patients and their caregivers, quality of patient–clinician communication and caregiver burden [7]. Of these, two self-reported measures, the 19-item Quality of Communication (QOC) questionnaire [8] and the 4-item ACP-Engagement (4-item ACP-E) questionnaire [6] were not available in Italian. We translated-adapted these two questionnaires. Moreover, to assess the communication skills of the physician involved in the ACP conversation considering the perspective of all the participants–the patient, the physician and, when applicable, the patient's significant other (SO), we devised a SO version (QOC-SO) and a physician version (QOC-Doc, not presented here) from the Italian version of the patient self-assessed QOC [7].

The objective of the present study was to culturally translate and adapt into Italian the 19-item QOC and the 4-item ACP-E questionnaires.

## Materials and methods

The study protocol was approved by the Ethics Committees of the Fondazione IRCCS Istituto Neurologico Carlo Besta, Milan (FINCB; clearance number: 83/2021) and the Azienda USL—IRCCS di Reggio Emilia (clearance number: 2021/0080829). All subjects gave their written consent and all procedures followed the Declaration of Helsinki.

### The questionnaires

Developed from qualitative studies with patients and clinicians, the QOC belongs to the ACP 'quality of care' domain [9]. The questionnaire (version 1.0) gauges the communication competences of the physician, and is interviewer-administered. This initial version of the QOC consists of 17 items measuring general communication (9 items) and communication about EOL care (8 items); included are also two items providing an assessment of the physicians' overall communication skills [8, 10]. Each item is rated on a scale from 0 ('very worst I can imagine'/ 'not at all') to 10 ('very best I can imagine'/ 'extremely'). Additional response items include "doctor didn't do" or "don't know". If the respondent endorses "doctor didn't do", the item is assigned a value of "0". This assignment was based on the assumption that, because all of the items identified important aspects of EOL communication, the failure to complete or address an item warranted a low score [11]. Two QOC scores are obtained by summing item responses, the range of possible scores being 0 (lowest skills) to 60 (highest skills) for 'general communication skills' (items 1, 2, 4, 5, 6, 7), and 0 to 70 for 'communication about EOL care' (items 8, 9, 10, 11, 13, 14, 16) [8].

Originally developed as an 82-item (50-minutes administration time) questionnaire measuring the complex behavior of ACP, the ACP-E is available in shorter versions (55-item, 34-item, 9-item, 4-item) [12]. The shorter versions worked well in a cohort of 986 English- and Spanish-speaking old adults from a US county hospital, and were able to detect within- and between-group changes comparable with the 82-item version [13]. Specifically, we were interested in the 4-item version, which assesses the readiness behavior change construct within the ACP 'process' domain [12]. The 4-item ACP-E responses are on a 5-point Likert scale: (1) 'I have never thought about it'; (2) 'I have thought about it, but I am not ready to do it'; (3) 'I am thinking about doing it in the next 6 months'; (4) 'I am definitely planning to do it in the

next 30 days'; (5) 'I have already done it' [14]. The total score is the average of the four item responses, and ranges from 1 (lowest engagement) to 5 (highest engagement).

## Cross-cultural adaptation

Following the International Society for Pharmacoeconomics and Outcomes Research Translation-Cultural Adaptation (ISPOR TCA) Task Force guidelines [15], we cross-culturally adapted the two questionnaires in five subsequent steps:

1. *Forward translation*: two qualified translators, both living in Italy, produced two independent translations. A panel consisting of the translators, a palliative care physician (S.V.), a psychologist (A.G.), a neurologist (A.S.), an expert patient (P.K.), and a lay person (M.F.) reviewed the forward translations (meeting 1) and a reconciled version was produced. Besides the professional translators, all the panel members were fluent in English. The panel was established for over 10 years except for S.V. and P.K., who joined more recently; both had previous experience of translation-adaptation.

2. *Backward translation*: the reconciled version generated in step 1 was independently translated back into US English by a third qualified translator, living in Italy. The backward translation was produced without access to the original version and without consulting the other translators.

3. *Pre-final version*: in a meeting (meeting 2) between those participating in step 1 and the backward translator, the backward translation was compared with the original, and further refinements to the Italian version were made. Differences were resolved by consensus, and a pre-final version was agreed.

4. *Expert feedback*: The pre-final version was read by an Italian researcher and clinical bioethicist (L.D.P.) who provided the translation panel comments and feedback on its coherency with the Law 219/2017. Finally, feedback was obtained from each questionnaire's authors. They received the translation grid, the backward translation produced by the translation panel, and were asked to compare the original questionnaire with the backward translation to identify any critical issues. The authors also received specific queries for items or instructions with problematic wording or conceptual ambiguities identified by the panel.

5. *Questionnaire refinement and devise of the QOC-SO*: Each translated questionnaire was refined after the expert feedback, and proof read. A patient self-assessed and a SO version (QOC-SO) were produced from the interviewer-administered Italian version, as well as a physician version (QOC-Doc), the latter including only the last two items (items 18 and 19), assessing the overall communication skills of the physician [7].

**Organization and documentation.** The Unit of Neuroepidemiology at FINCB had responsibility for the translation-cultural adaptation methodology, devised the materials and procedures, asked permission and involved the questionnaire's authors, and oversaw each stage of the process. Meetings 1 and 2 were held online (Teams conference system) and recorded. The original questionnaires are available at the University of Washington School of Medicine website [11], and the University of California 'Prepare for your Care' website [14]. The whole process is reported in a translation grid (S1 and S2 Files), which was available to each member of the panel before each meeting to facilitate discussion on the semantic, conceptual, and normative equivalence of the questionnaire introduction, items and response options. Challenging phrases, uncertainties and rationale of final decisions are reported in the

translation grid. The grid also contains queries sent to the questionnaire's authors, and their responses. After meeting 2, the translation grid was reviewed by each panel member and by the scale authors for validation.

## Cognitive debriefing

**Eligibility criteria.** Participants (i.e., patients and SOs) were enrolled at three palliative care centers with inpatient, outpatient and home-based palliative care facilities; two centers are in Northern Italy (Reggio Emilia and Turin) and one in Central Italy (Spoleto).

Participants were selected using a purposeful sampling technique ensuring diversity in age and education. They were adults (age ≥18 years), fluent in Italian, and had provided written informed consent to participate in the study. Patients with severe cognitive compromise (clinical judgment), and those with impairments precluding communication were excluded. For each questionnaire, we pre-planned a minimum of five interviews according to Willis' indications [16], and the modified Tourangeau model of cognitive aspects [17]. Patients debriefed the QOC questionnaire (interview or self-assessed version) and the 4-item ACP-E questionnaire. SOs debriefed the QOC-SO.

**Procedure.** The referring physician: a) informed the participant about the study and provided the informed consent form; b) confirmed that all the eligibility criteria were met; c) recorded on the clinical record form of the consenting patient the following information: gender, age, education, and current occupation. The interviewer recorded the following SO information: gender, age, education, current occupation, and relationship with the patient.

The interviews were face-to-face, via videoconference or on the phone based on participant's preference. The interviewers (L.D.P., M.P., M.C.) used an interview guide previously drawn up and agreed by the study authors (S3 File); they took written notes (interviews were not recorded). The interviewer checked that all the eligibility criteria were satisfied. She invited the participant to complete the self-assessed questionnaires alone, or administered the QOC interviewer version. There followed a series of open-ended questions to explore the interviewee's understanding of each questionnaire as a whole and considering each item, and response options. The interviewees were invited to offer alternative words or paraphrase statements, and they were asked about the questionnaire's acceptability (length, layout, readability). The interviewers did not have any existing relationship with the participants.

**Organization and documentation.** The Bioethics Unit, Azienda USL—IRCCS di Reggio Emilia had responsibility for the qualitative study, devised the interview guides, trained the interviewers, and performed the qualitative analysis. Participants were recruited from three centers: the Palliative Care Unit, Azienda USL—IRCCS di Reggio Emilia; the Fondazione FARO, Turin; and the Hospice "La Torre sul Colle", USL Umbria 2, Spoleto. Each center had responsibility for participant's screening and enrollment, and recorded the general and clinical information.

## Analysis

Continuous data were summarized using medians and ranges, while categorical data were described as numbers and frequencies.

Interview notes were reviewed independently by two researchers (L.D.P., R.M.Z.) using content analysis to identify areas of misunderstanding, and where modifications to wording or layout were indicated [18, 19]. The two reports were compared and discussed jointly by the two researchers, who produced a final report. We followed the consolidated criteria for reporting qualitative research (COREQ) [20]. S4 File reports the COREQ checklist for the current study.

## Results

### Cross-cultural adaptation

QOC: The translation-cultural adaptation process of the QOC is summarized in Table 1, and detailed in S3 File. The pre-final Italian version of the questionnaire, as well as the patient self-assessed and the SO version (QOC-SO) devised from it, are available at the FINCB website [21].

**Introduction/instructions.** The translation of this section went smoothly.

**Items.** Fourteen of the 19 items (items 2, 5–7, 9–12, 14–19) were easily forward and back-translated, and comparison of the back translation with the original confirmed the equivalence of the Italian version with the original scale. As for the other items, the panel had problems regarding consistency in the wording (4 items) and semantic equivalence (one item). Specifically, in the original questionnaire, the doctor is named as 'doctor [X]' (introduction to item 1), 'this doctor' (introduction to items 18 and 19; item 19), and 'your doctor' (item 18). To increase consistency, in the pre-final Italian translation we used *il suo medico* ('your doctor') in all instances. The term 'treatment' (*terapie*) was present in items 3, 4 and 13 of the original questionnaire, while in the introduction the term was 'medical care' (*assistenza*—which is more comprehensive than *terapie*). The scale author confirmed that it was OK to have *assistenza* in the introduction and *terapie* in items 3/4/13. Lastly, the translation of the word 'feelings' was difficult to the panel. After the author clarified that the focus was the emotional component of the discussion, the panel agreed on *pensieri* (thoughts/worries).

**Response options.** No difficulties were found, except for the response option 'didn't do', which the panel considered confounding and coincident with the '0' score ('The very worst I could imagine') on the numeric scale. The answer 'not applicable' was considered as viable in case the behavior could not be assessed (e.g., on item 2, 'Looking you in the eye', by a visually impaired person or in a telephone consultation; on item 3, 'Including your loved ones in decisions about your illness and treatment', in a consultation where the patient is alone). Thus, the panel agreed to skip the response option 'didn't do', and to add *non valutabile* ('not applicable').

**Layout/format.** The only revision was made to adapt the text to the administration format of the questionnaire, and concerned the expression 'please select the best number for each statement' (introduction, instruction on items 18 and 19). This expression was translated as *selezioni* ('select') in the electronic, and *segni* ('tick') in the paper format.

4-item ACP-E: The translation-cultural adaptation process of the 4-item ACP-E is summarized in Table 2, and detailed in S2 File. The pre-final Italian versions (patient self-assessed, and interviewer versions) are available at the FINCB website [22].

**Table 1. Overview of revisions made to the 19-item Quality of Communication questionnaire.**

| Original content | Revised content | Reason for change |
|---|---|---|
| 'Doctor [X]' (introduction to item 1), 'this doctor' (introduction to items 18 and 19; item 19), and 'your doctor' (item 18) | *Il suo medico* ('your doctor') in all instances | To improve clarity and internal consistency |
| 'Feelings' (item 8) | *Pensieri* (thoughts/worries) | To improve clarity |
| 'Didn't do' (response option) | 'Not applicable' | To improve consistency with the instructions |
| 'Please select the best number for each statement' (introduction, instruction to items 18 and 19) | *Segni* ('tick') in the paper, and *selezioni* ('select') in the electronic format | To adapt to the format (paper or electronic) |

**Table 2. Overview of revisions made to the 4-item ACP engagement questionnaire.**

| Original content | Revised content | Reason for change |
|---|---|---|
| 'Medical decision makers, or surrogates' (introduction/instructions) | *'Fiduciario'* | To improve coherence with the Italian ACP legislation |
| 'Medical treatments' (*trattamenti sanitari*) is used in the introduction to topic 2, while 'medical care' (*assistenza*–a broader term in Italian) is used in questions 2 and 4. | *'Assistenza'* | To improve clarity and internal consistency |
| Introduction/instructions | The following sentence has been added: 'For each statement, select the answer that describes at best your current situation' (*per ciascuna domanda, scelga la risposta che meglio descrive la sua situazione attuale*). | To improve clarity |
| Titles, sub-titles, numbering (of topics and items) | Removal of contents not pertinent to the patient's readiness domain | To simplify the layout |

**Introduction/instructions.** The translators had difficulties in translating 'medical decision makers, or surrogates'. Following the bioethicist's advice, the panel agreed that, consistently with the Law 219/2017, one specific Italian term should be used: *fiduciario*. This was considered acceptable in terms of cognitive demand as this technical term is explained thereafter in the questionnaire: '[. . .] a family member or friend who can make decisions for you if you were to become too sick to make your own decisions'. Another challenge was that, in the introduction to topic 2 (original questionnaire) the expression 'medical treatments' (*trattamenti sanitari*) is used, while in questions 2 and 4 it is 'medical care' (*assistenza*–a broader term in Italian). A query was generated to the ACP-E author, who agreed on using the broader expression consistently along the questionnaire. In addition, the panel made two main revisions to this section. First, the sentence 'Please try to answer as honestly as you can' and 'Please give us your honest opinions' were considered conceptually equivalent, and reported only once in the Italian version (repetition can be appropriate in longer versions of the ACP-E, see Layout paragraph below). Second, we added the following instruction at the end of this section: 'For each statement, select the answer that describes at best your current situation' (*per ciascuna domanda, scelga la risposta che meglio descrive la sua situazione attuale*). These changes were agreed on by the ACP-E author.

**Questions.** No difficulties were found, and the comparison of the back translation with the original confirmed the equivalence of the Italian 4-item ACP-E with the original scale. As a single remark concerning coherence with the Italian Law 219/2017, on item 1, the expression 'one or more people' was preferred to 'a person or group of people'. In fact, in the Italian Law, when more than one medical decision maker is nominated by the patient, they are ordered and act individually, not as a group; if the first person on the list is not available, the second one is contacted, and so on.

**Response options.** No difficulties were found on any response option.

**Layout.** The layout and structure of the 4-item ACP-E is the same of the longer questionnaire versions. However, the panel noted that, in the shortest (4-item) version, this structure is disproportionate to the questionnaire contents [14]. Titles, sub-titles, numbering (of topics and items) increase the questionnaire complexity, particularly as the 4-item ACP-E focuses on the patient's readiness domain only. Thus, the panel proposed a simplified layout in the Italian version [22]. These changes were agreed on by the ACP-E authors.

## Cognitive debriefing

Between October 2021 and June 2022, a total of 14 patients and four SOs agreed to participate in the study. Of these, four patients and one SO did not participate due to personal or organizational issues. One patient left the study before receiving the interview (see below). The characteristics of participants are reported in Table 3. Six interviews were held by M.P., three by M.C., and three by L.D.P. The interview guides are reported in S3 File. The interviews lasted between 26 and 60 minutes.

QOC and QOC-SO: We devised a QOC-SO from the pre-final Italian QOC, to be used in both interview and self-assessed administration [21]. After the first few interviews, it emerged that the items on physician's communication about EOL care (items 10–17) and the item on physician's overall EOL communication skills (item 18) were taxing to both patients and SOs. Specifically, one patient (Co005) did not complete items 10–18, and asked to re-schedule the interview; when contacted, she declined participation. One patient (Co004) completed the first seven items and then asked the interviewer to administer him the remaining. One patient and one SO completed the whole questionnaire, however during the interview they recommended a 'researcher-assisted completion' (Table 4). After discussion by the Steering Committee and Data Safety and Monitoring Committee of the ConCure-SM project, it was decided to complete the debriefing by using the interview version of the QOC (and the QOC-SO administered by the interviewer). In addition, this finding determined a major amendment in the protocol

**Table 3. General and clinical data of participants in the cognitive debriefing.** One of the nine patients was not interviewed (she left the study after partial completion of the Quality of Communication questionnaire).

| | Patients (n = 9) | Significant others (n = 3) |
|---|---|---|
| **Characteristic** | *No (%)* | |
| Age, years* | 69 (42–89) | 38 (35–70) |
| Women | 4 (44%) | 1 (33%) |
| Education (degree) | | |
| Middle school | 3 (33%) | 0 (0%) |
| High school | 3 (33%) | 2 (67%) |
| University | 3 (33%) | 1 (33%) |
| Working status | | |
| Retired | 6 (66%) | 1 (33%) |
| Occupied (public officer, veterinary) | 2 (22%) | 2 (67%) |
| Unemployed | 1 (11%) | 0 (0%) |
| Administration of the interview | | |
| Face to face | 7 (78%) | 0 (0%) |
| Online | 1 (11%) | 2 (67%) |
| On the phone | 1 (11%) | 1 (33%) |
| Main diagnosis | | |
| Cancer (kidney n = 2, rectum, bladder, liver, lung, prostate) | 7 (78%) | – |
| Heart failure | 1 (11%) | – |
| Echinococcosis | 1 (11%) | – |
| Disease duration, months* | 36 (7–72) | – |
| Relation with the patient | | |
| Son | – | 2 (67%) |
| Friend and trustee | – | 1 (33%) |

* Median (min-max)

**Table 4. Main findings of the cognitive debriefing of the Quality of Communication scale.** SO, significant other.

| Content | Where in the scale | Participant |
|---|---|---|
| Focusing on a specific physician can be challenging, particularly when followed in hospice or by a team | Introduction: 'the doctor taking care of your condition' | Co007 |
| | | Co012 |
| | | Co015_SO |
| | | Co016_SO Co017_SO |
| Looking into patient's eyes is a key element of communication | Item 2 | Co013 |
| At first unexpected, then obvious issue (i.e. looking into patient's eyes) | Item 2 | Co014 |
| Statement difficult to understand, requires attentive reading | Item 8 | Co010 |
| Same content, item 9 could be skipped | Item 8 vs. 9 | Co007 Co014 |
| Content could be confused in the patient version | Item 8 vs. 9 | Co015_SO Co016_SO |
| 'Aspettativa di vita' difficult to understand | Item 10 | Co012 Co014 |
| Statement understandable after re-reading | Item 13 | Co010 |
| Patient's values and beliefs are not part of the patient-physician relationship, except for a long-lasting one | Items 14–17 | Co014 |
| The physician is not expected to be competent on these issues, to be deserved e.g. to a psychologist or to a chaplain | Items 14–17 | Co004 Co012 Co014 |
| Patients encounter many physicians in their disease trajectory | Item 19 | Co016_SO |

of the pilot trial (see Discussion). Both patients and SOs considered the questionnaire overall clear and understandable with some exceptions. They judged the response options clear and well differentiated, including the options 'non saprei' and 'non valutabile'. They found the topic of patient-physician communication very important, including EOL communication. However, both types of respondents reported that the questionnaire's contents were emotionally taxing. This was the case particularly for items 8–13, and 18 (Table 4). One patient initially found item 2 ('looking into patient's eyes') as 'strange' and then 'obvious'. Another patient appreciated this item, as an indicator of a good communication style. Two patients considered item 8 a bit difficult (one of them had to read again the item to fully get it). One patient considered items on patient's values and beliefs not pertinent to the medical encounter, out of a long-lasting relation. Finally, three patients said that the physician was not expected to be competent on these issues.

4-item ACP-E: Patients considered the survey clear and understandable, and useful in the (advance) care process. They viewed the layout as adequate: repetitions, use of bold and underlined text, and large font size helped identification of the sessions, and brevity of the survey a plus. Patients recommended replacing self-completion with an 'assisted administration' for two main reasons: complexity related to ACP, and dealing with emotions. Concerning complexity, one patient (Co010) considered the introduction a bit difficult, including the phrase 'there are no right or wrong answers'. The word 'fiduciario' was new to two patients (Co008, Co004), and both asked for additional explanation to the description reported in the body of the survey. One patient (Co007) who previously signed her advance directive document found the survey clear and easy to complete. She suggested to skip the English title of the survey.

## Discussion

This paper reports the cross-cultural adaptation into Italian of two questionnaires addressing the ACP process, both developed in the US. Our adaptation framework was the ISPOR TCA

Task Force guideline [15]. In order to ensure that the Italian version of the QOC and 4-item ACP-E are sensitive to the local context and norms while remaining equivalent to the original measure, we involved the questionnaire's authors, as well as an Italian clinical bioethicist. The panel proposed some changes in order to simplify the layout and structure of the 4-item ACP-E, which focuses on the ACP patient's readiness domain only. The cognitive debriefing of the QOC highlighted ways in which the QOC's EOL communication items were found to be challenging, to both patients and SOs. To protect participants in the ConCure-SM pilot trial from the challenges of these items, we decided to make a major amendment to the trial protocol [7]. The amendment consisted in the use of the interviewer-administered version of the QOC/QOC-SO in an adaptive form: the section on communication about EOL care (8 items), and one item on the overall (EOL) communication skills of the physician are administered only to participants in whom this topic was addressed during the ACP conversation. We acknowledge that patient-clinician discordance on the occurrence of EOL discussions has been reported [23], and will further explore the acceptability of EOL items in the subsequent psychometric testing phase. We also acknowledge that cultural issues arose during the study. These include a lack of knowledge of the Italian ACP legislation, revealed by the fact that some respondents were unaware of the term '*fiduciario*'. Another problem is the difficulty of talking about EOL in the Italian clinical context: although all enrolled patients suffered from progressive conditions, they expected their doctors to talk about possible treatments and care pathways, not about EOL. Death being a taboo topic, along with the superstitious belief that talking about something can evoke it, may contribute to this difficulty.

Publications on cross-cultural adaptation typically focus on the psychometric (i.e., statistical) properties of the instrument. The translation of self-reported outcome measures in the target culture are generally poorly documented, despite the fact that these are key components of instrument's validity, and a prerequisite of measurement equivalence [15, 24, 25]. For this reason, and consistently with our recent policy [26, 27], we described the translation-cultural adaptation and cognitive debriefing phases in a dedicated paper. Important issues have emerged regarding the QOC, informing the development of an interview as well as a self-assessed version of the instrument. Concerning the 4-item ACP, we believe that the simplified layout can ease its administration and the assessment of ACP readiness in research and in routine patient care.

This study has some limitations. Specifically, we interviewed only three SOs, and data saturation was not discussed (S4 File).

## Conclusions

The Italian adaptation of the QOC and 4-item ACP-E questionnaires were satisfactory in terms of semantic, conceptual and normative equivalence. Acceptability was satisfactory for the 4-item ACP-E, while the items on EOL of the QOC were emotionally taxing, suggesting the use of the interviewer version of the instrument. Psychometric testing of both questionnaires on a large, independent sample will follow.

## Supporting information

**S1 File. Translation grid QOC.**
(PDF)

**S2 File. Translation grid 4-ACP-E.**
(PDF)

**S3 File. Interview guide.**
(PDF)

**S4 File. COREQ checklist.**
(PDF)

## Acknowledgments

We are indebted to the patients and the significant others who provided their valuable input.

**ConCure-SM project investigators**

**Steering Committee:** M. Cascioli, L. De Panfilis, M.G. Grasso, A. Giordano, A. Lugaresi, E. Pucci (UOC Neurologia, ASUR Marche, Fermo, Italy), A. Solari, C. Solaro, S. Veronese, M. Bruzzone (The Italian Multiple Sclerosis Society, Genoa, Italy), P. Kruger, A. Gajofatto, F. Patti.

**Data Safety and Monitoring Committee:** K. Brazil (School of Nursing and Midwifery, Queen's University of Belfast, Belfast, Northern Ireland, UK), B. Farsides (Brighton & Sussex Medical School, Falmer, Brighton, UK), L. Orsi (The Italian Society of Palliative Care (SICP), Milan, Italy;), C. Peruselli (SICP, Milan, Italy), and D. Oliver (The Tizard Centre, University of Kent, Canterbury, UK) (Chair).

**Data Management and Analysis Committee:** M. Farinotti (data manager), and A. Giordano.

**Qualitative Analysis Panel:** M. Cascioli, L. De Panfilis, L. Ghirotto (Qualitative Research Unit, Azienda USL-IRCCS di Reggio Emilia, Reggio Emilia, Italy), K. Mattarozzi (Department of Experimental, Diagnostic and Specialistic Medicine, School of Medicine, Alma Mater Studiorum University of Bologna, Italy), M. Perin, and S. Veronese.

**Health Professional Training Panel:** M. Cascioli, L. De Panfilis, K. Mattarozzi, E. Pucci, M. Rimondini (Section of Clinical Psychology, Department of Neuroscience, Biomedicine and Movement Sciences, University of Verona, Policlinico G.B. Rossi, Verona, Italy;), A. Solari, and S. Veronese.

**Linguistic validation Panel:** M. Farinotti, A. Giordano, P. Kruger, A. Solari, S. Veronese and three independent translators.

**Enrolling Centers:** *Department of Neuroscience, Biomedicine and Movement Sciences, University of Verona; Unit of Neurology, Borgo Roma Hospital, Azienda Ospedaliera Universitaria Integrata Verona*: A. Gajofatto, F. Gobbin, R. Orlandi; *Department of Rehabilitation M. L. Novarese Hospital, Moncrivello, Vercelli*: C. Solaro, E. Grange; *Azienda USL-IRCCS di Reggio Emilia, Reggio Emilia*: L. De Panfilis, S. Montepietra, F. Sireci; *UOSI Riabilitazione Sclerosi Multipla, IRCCS Istituto delle Scienze Neurologiche di Bologna; Dipartimento di Scienze Biomediche e Neuromotorie, Università di Bologna, Bologna*: A. Lugaresi, L. Sabbatini, C. Scandellari, E. Ferriani; *Fondazione IRCCS Santa Lucia, Roma*: M.G. Grasso, G. Presicce; *University Hospital Policlinico Vittorio Emanuele, Catania*: F. Patti, C.G. Chisari, S. Toscano.

## Author Contributions

**Conceptualization:** Alessandra Solari.

**Data curation:** Ludovica De Panfilis, Marta Perin, Marta Cascioli, Mariangela Farinotti, Roberta M. Zagarella, Alessandra Solari.

**Formal analysis:** Alessandra Solari.

**Funding acquisition:** Alessandra Solari.

**Methodology:** Alessandra Solari.

**Supervision:** Alessandra Solari.

**Validation:** Ludovica De Panfilis, Simone Veronese, Mariangela Farinotti, Paola Kruger, J. R. Curtis, Rebecca L. Sudore, Elizabeth L. Nielsen, Ruth A. Engelberg, Andrea Giordano, Alessandra Solari.

**Visualization:** Mariangela Farinotti, Alessandra Solari.

**Writing – original draft:** Alessandra Solari.

**Writing – review & editing:** Ludovica De Panfilis, Simone Veronese, Marta Perin, Marta Cascioli, Mariangela Farinotti, Paola Kruger, Roberta M. Zagarella, J. R. Curtis, Rebecca L. Sudore, Elizabeth L. Nielsen, Ruth A. Engelberg, Andrea Giordano, Alessandra Solari.

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
