## [Decision Letter · Decision Letter 0]

1 Feb 2023

PONE-D-22-30497Italian cross-cultural adaptation of the Quality of Communication questionnaire and the 4-item Advance Care Planning Engagement questionnairePLOS ONE

Dear Dr. Giordano,

Thank you for submitting your manuscript to PLOS ONE. After careful consideration, we feel that it has merit but does not fully meet PLOS ONE’s publication criteria as it currently stands. Therefore, we invite you to submit a revised version of the manuscript that addresses the points raised during the review process.

We look forward to receiving your revised manuscript.

Kind regards,

Francesca Baratta, PharmD, PhD

Academic Editor

PLOS ONE

Journal Requirements:

"This research was funded by Fondazione Italiana Sclerosi Multipla (FISM; aism.fism.it), grant no. 2020/R-Multi/024 to A.S, and partially supported by the Italian Ministry of Health (RRC). R.L.S. is funded in part by the National Institute on Aging, National Institutes of Health (K24AG054415)."             

"A.S. reports grants from the Italian Multiple Sclerosis Foundation (FISM) and the European Academy of Neurology, during the conduct of the study; she serves as board member for Merck Serono, and received personal fees from Almirall and Merck Serono, outside the submitted work."

Reviewers' comments:

Reviewer's Responses to Questions

**Comments to the Author**

1. Is the manuscript technically sound, and do the data support the conclusions?

Reviewer #1: Yes

Reviewer #2: No

Reviewer #3: Yes

2. Has the statistical analysis been performed appropriately and rigorously? 

Reviewer #1: N/A

Reviewer #2: Yes

Reviewer #3: Yes

3. Have the authors made all data underlying the findings in their manuscript fully available?

Reviewer #1: No

Reviewer #2: Yes

Reviewer #3: Yes

4. Is the manuscript presented in an intelligible fashion and written in standard English?

Reviewer #1: Yes

Reviewer #2: Yes

Reviewer #3: Yes

5. Review Comments to the Author

Reviewer #1: At the end of your introduction, state the objective of your research instead of what is written in the last paragraph “ For each questionnaire, the linguistic validation encompasses two main actions: translation97 adaptation of the QOC and ACP-E inventories into Italian; psychometric validation. Here we report the results of the first action”.

The main aspect of the paper that can improve is the presentation of the cross-cultural adaptation (Results from page 9-12). Is it possible to create Tables to explain: what items were modified, why and how?

The conclusion should summarise your main results. Avoid citing references in your conclusion.

Reviewer #2: The authors translated and cross-culturally validated the Italian the Quality of Communication (QOC) and the 4-item ACP Engagement (4-item ACP-E). Their results in 14 subjects showed satisfactory semantic, conceptual and normative equivalence. Below my comments.

The authors report that “For each questionnaire, the linguistic validation encompasses two main actions: translation adaptation of the QOC and ACP-E inventories into Italian; psychometric validation”. But they claim to report only the first part (i.e. Translation and adaptation). Why did the authors decide not to report the results of the psychometric properties? Even if the questionnaire has been transculturally translated, if we do not know the psychometric properties (such as reliability and validity) we cannot use it with our patients and therefore the repercussions in clinical practice are very low. I highly recommend reporting the psychometric results as well. Furthermore, I have hardly read of papers reporting only the assessment of cross-cultural validity; these results are usually reported together with the validity and reliability results, to improve the application of the results in clinical practice.

The authors report that “the ACP-E is available in shorter versions (55-item, 34-item, 9-item, 4-item) … Specifically, we were interested in the 4-item version”. What are the reasons that led the authors to prefer the 4-item version? Why didn't they consider translating the entire completed questionnaire?

Information on the structure of the questionnaire is missing. For example, there is no information about the scores for each item and the calculation of the total score

The authors included only 14 patients in the cognitive de briefing. Typically, guidelines for cross-cultural validation suggest including 30 patients. Why did the authors only include 14 of them?

Reviewer #3: Peer-Review PONE-D-22-30497

Dear authors,

Thank you for submitting your manuscript to the journal PLOS ONE. This is a well-prepared and thorough investigation of the content validity of the Italian translations of two questionnaire relevant for supporting Advance Care Planning. I only have very minor comments below.

Clinical relevance of the questionnaires

In the introduction, you explain about ACP - what it is, its elements etc. You also specify that the two questionnaires be used for clinical and research purposes (page 4, line 82/83). I would recommend being a bit more specific regarding the use cases you envisage. Are you translating the questionnaires for evaluation purposes of ACP interventions? Or are you planning to bring them into clinical practice? If yes, in which setting? How is shared decision making going to happen with the aid of these questionnaires?

You can also specify this in the discussion section, not in the background section if you felt it would sit better there.

Methods

You are very comprehensive when it comes to explaining your methods. Although you specify the methods for cognitive interviewing and you also specify the analysis methods, I was missing the framework of Tourangeau being applied to the data. From the results and how they are reported, I would expect you probably used Tourangeau. It might be worth mentioning him and his framework.

Discussion/article

Your aim is the cross-cultural adaptation of the two ACP questionnaires. You do a very thorough translation process. However, although cultural issues are implicitly mentioned in the results, the discussion section then does not make them explicit. You present a lot of evidence in the background on the Italian's readiness to endorse ACP. You also describe the health care system's lack in responding to this need. You describe the "emotionally taxing" nature of some of the items. I was wondering whether it would be possible to explicitly discuss cultural issues and how they reflect on the construct ACP the questionnaire want to measure. After all, achieving translations that are faithful to the original definition and measurement of the construct is one of the aims of a cross-cultural adaptation. The other might be to learn more about once culture and, in reflection, about the construct under investigation. Results from cross-cultural adaptations of questionnaires can feed back into further develop the content validity of the questionnaires. I was missing a section in the discussion to do just that - a bit of reflection on the general content validity of the two questionnaires.

Minor comment

In the introduction section, you often speak of "his/her own personal values" or "his/her health". It is common to do a gender-neutral pronoun in plural in English in these situations (at least in publications, not in the actual questionnaires). Page 4, line 66 could better read as "ACP helps the patient to identify their own personal values and goals, understand their health..."

Otherwise, this was an engaging, clearly structured and methodologically thoroughly prepared study. What a pleasure to read!

6. PLOS authors have the option to publish the peer review history of their article (what does this mean?). If published, this will include your full peer review and any attached files.

Reviewer #1: No

Reviewer #2: **Yes: **Leonardo Pellicciari

Reviewer #3: **Yes: **Christina Ramsenthaler

---

## [Author Response · Author response to Decision Letter 0]

6 Feb 2023

Reviewer #1:

At the end of your introduction, state the objective of your research instead of what is written in the last paragraph “For each questionnaire, the linguistic validation encompasses two main actions: translation adaptation of the QOC and ACP-E inventories into Italian; psychometric validation. Here we report the results of the first action”.

R: We have revised the last sentence of the introduction as suggested.

The main aspect of the paper that can improve is the presentation of the cross-cultural adaptation (Results from page 9-12). Is it possible to create Tables to explain: what items were modified, why and how?

R: We have added two tables to summarize the changes made in the QOC (Table 1) and 4-item ACP-E (Table 2).

The conclusion should summarise your main results. Avoid citing references in your conclusion.

R: We have revised the conclusion as suggested.

Reviewer #2: 

The authors report that “For each questionnaire, the linguistic validation encompasses two main actions: translation adaptation of the QOC and ACP-E inventories into Italian; psychometric validation”. But they claim to report only the first part (i.e. Translation and adaptation). Why did the authors decide not to report the results of the psychometric properties? 

R: We purposely focused on a research phase that is usually poorly documented: translation and cultural adaptation (and cognitive debriefing). We have aired this in the discussion section. 

Even if the questionnaire has been transculturally translated, if we do not know the psychometric properties (such as reliability and validity) we cannot use it with our patients and therefore the repercussions in clinical practice are very low. I highly recommend reporting the psychometric results as well. Furthermore, I have hardly read of papers reporting only the assessment of cross-cultural validity; these results are usually reported together with the validity and reliability results, to improve the application of the results in clinical practice.

R: The psychometric testing phase is currently ongoing and will be reported in a separate manuscript. 

The authors report that “the ACP-E is available in shorter versions (55-item, 34-item, 9-item, 4-item) … Specifically, we were interested in the 4-item version”. What are the reasons that led the authors to prefer the 4-item version? Why didn't they consider translating the entire completed questionnaire?

R: We were interested in the 4-item version of the 4-item ACP-E as: (a) it had good sensitivity to change compared to longer versions in a large trial involving US English- and Spanish-speaking old adults (PREPARE study, ref. 12); (b) it assesses the readiness behavior change construct within the ACP ‘process’ domain (ref. 11); (c) administration burden is limited. We have described this in the materials and methods section. 

Information on the structure of the questionnaire is missing. For example, there is no information about the scores for each item and the calculation of the total score

R: Following reviewer’s advice, we have added (methods section) the scoring rules for the two scales. 

The authors included only 14 patients in the cognitive de briefing. Typically, guidelines for cross-cultural validation suggest including 30 patients. Why did the authors only include 14 of them?

R: As from the ISPOR Task Force, we applied cognitive debriefing, which involves a small group of 5-6 subjects. We guess that the reviewer refers to pilot testing instead, which requires 30–40 subjects.

Reviewer #3: Peer-Review PONE-D-22-30497

Clinical relevance of the questionnaires

In the introduction, you explain about ACP - what it is, its elements etc. You also specify that the two questionnaires be used for clinical and research purposes (page 4, line 82/83). I would recommend being a bit more specific regarding the use cases you envisage. Are you translating the questionnaires for evaluation purposes of ACP interventions? Or are you planning to bring them into clinical practice? If yes, in which setting? How is shared decision making going to happen with the aid of these questionnaires?

You can also specify this in the discussion section, not in the background section if you felt it would sit better there.

R: We have described (introduction section) that we are using the QOC and the 4-item ACP-E for evaluation purposes of an ACP intervention (ConCure-SM). Following reviewer’s advice, we have expanded the description of the ConCure-SM intervention and feasibility trial to clarify the context in which the two instruments are being used. 

Methods

You are very comprehensive when it comes to explaining your methods. Although you specify the methods for cognitive interviewing and you also specify the analysis methods, I was missing the framework of Tourangeau being applied to the data. From the results and how they are reported, I would expect you probably used Tourangeau. It might be worth mentioning him and his framework.

R: We thank the reviewer for her comment. We do confirm that we pre-planned the interviews according to Willis' indications [14], and the modified Tourangeau model of cognitive aspects [15]. We have now slightly changed the text, accordingly.

Discussion/article

Your aim is the cross-cultural adaptation of the two ACP questionnaires. You do a very thorough translation process. However, although cultural issues are implicitly mentioned in the results, the discussion section then does not make them explicit. You present a lot of evidence in the background on the Italian's readiness to endorse ACP. You also describe the health care system's lack in responding to this need. You describe the "emotionally taxing" nature of some of the items. I was wondering whether it would be possible to explicitly discuss cultural issues and how they reflect on the construct ACP the questionnaire want to measure. After all, achieving translations that are faithful to the original definition and measurement of the construct is one of the aims of a cross-cultural adaptation. The other might be to learn more about once culture and, in reflection, about the construct under investigation. Results from cross-cultural adaptations of questionnaires can feed back into further develop the content validity of the questionnaires. I was missing a section in the discussion to do just that - a bit of reflection on the general content validity of the two questionnaires.

R: In the discussion section, we have now made explicit the main cultural issues that emerged during the cross-cultural adaptation process: “We also acknowledge that cultural issues arose during the study. These include a lack of knowledge of the Italian ACP legislation, revealed by the fact that some respondents were unaware of the term 'fiduciario'. Another problem is the difficulty of talking about EOL in the Italian clinical context: although all enrolled patients suffered from progressive conditions, they expected their doctors to talk about possible treatments and care pathways, not about EOL. Death being a taboo topic, along with the superstitious belief that talking about something can evoke it, may contribute to this difficulty.”

Minor comment

In the introduction section, you often speak of "his/her own personal values" or "his/her health". It is common to do a gender-neutral pronoun in plural in English in these situations (at least in publications, not in the actual questionnaires). Page 4, line 66 could better read as "ACP helps the patient to identify their own personal values and goals, understand their health..."

R: We have changed the sentence as suggested.

Otherwise, this was an engaging, clearly structured and methodologically thoroughly prepared study. What a pleasure to read!

R: We thank the reviewer for her appreciation of the study.

---

## [Decision Letter · Decision Letter 1]

28 Feb 2023

Italian cross-cultural adaptation of the Quality of Communication questionnaire and the 4-item Advance Care Planning Engagement questionnaire

PONE-D-22-30497R1

Dear Dr. Giordano,

We’re pleased to inform you that your manuscript has been judged scientifically suitable for publication and will be formally accepted for publication once it meets all outstanding technical requirements.

Kind regards,

Francesca Baratta, PharmD, PhD

Academic Editor

PLOS ONE

Reviewers' comments:

Reviewer's Responses to Questions

**Comments to the Author**

1. If the authors have adequately addressed your comments raised in a previous round of review and you feel that this manuscript is now acceptable for publication, you may indicate that here to bypass the “Comments to the Author” section, enter your conflict of interest statement in the “Confidential to Editor” section, and submit your "Accept" recommendation.

Reviewer #2: All comments have been addressed

Reviewer #3: All comments have been addressed

2. Is the manuscript technically sound, and do the data support the conclusions?

Reviewer #2: Yes

Reviewer #3: Yes

3. Has the statistical analysis been performed appropriately and rigorously? 

Reviewer #2: Yes

Reviewer #3: N/A

4. Have the authors made all data underlying the findings in their manuscript fully available?

Reviewer #2: Yes

Reviewer #3: Yes

5. Is the manuscript presented in an intelligible fashion and written in standard English?

Reviewer #2: Yes

Reviewer #3: Yes

6. Review Comments to the Author

Reviewer #2: I thank the authors for their efforts to answer my questions. I agree that the purpose of the paper is to describe the cross-cultural validation process. Unfortunately, in this form the results of this study have limited application in clinical practice. Having the validity and reliability data available allows clinicians to be able to use this tool. I hope these results will be available soon.

Reviewer #3: (No Response)

7. PLOS authors have the option to publish the peer review history of their article (what does this mean?). If published, this will include your full peer review and any attached files.

Reviewer #2: **Yes: **Leonardo Pellicciari

Reviewer #3: **Yes: **Christina Ramsenthaler

---

## [Editor Report · Acceptance letter]

15 Mar 2023

PONE-D-22-30497R1 

Italian cross-cultural adaptation of the Quality of Communication questionnaire and the 4-item Advance Care Planning Engagement questionnaire 

Dear Dr. Giordano:

I'm pleased to inform you that your manuscript has been deemed suitable for publication in PLOS ONE. Congratulations! Your manuscript is now with our production department. 

Kind regards, 

on behalf of

Dr. Francesca Baratta 

Academic Editor

PLOS ONE